# Gender distribution of Top Doctors in otolaryngology-head and neck surgery

**Lauren A. DiNardo**[1]*, **Alyssa D. Reese**[1], **Maya Raghavan**[1], **Meagan Sullivan**[1], **Michele M. Carr**[2]

**1** Jacobs School of Medicine and Biomedical Sciences at the University at Buffalo, Buffalo, New York, United States of America, **2** Department of Otolaryngology, Jacobs School of Medicine and Biomedical Sciences at the University at Buffalo, Buffalo, New York, United States of America

* ldinardo@buffalo.edu

**Data Availability Statement:** All data is publicly available on www.castleconnolly.com.

**Funding:** The author(s) received no specific funding for this work.

## Abstract

### Introduction

Our study seeks to understand the profiles of otolaryngologists selected by Castle Connolly's Top Doctor list and how this compares to the entire field of otolaryngology.

### Methods

Top Doctor lists published in Castle Connolly affiliated magazines were analyzed for Otolaryngology, Otolaryngology/Facial Plastic Surgery, or Pediatric Otolaryngology physicians. Only lists published in 2021 or representing the 2021 Top Doctor lists were analyzed. Of the total 39 partnered magazines, 27 met our criteria. Information on the physician was analyzed from the Castle Connolly website and included: gender, education, faculty position, years as a Top Doctor, and certifications of each physician.

### Results

879 doctors, 742 (84%) men and 137 women (16%), were included in our analysis. 509 physicians completed a fellowship, 85 (62%) women and 424 (57%) men. The fellowship type varied significantly between gender (p = .002). 122 (14%) Top Doctors completed facial and plastic reconstructive surgery and 111 (91%) were men. Of the women Top Doctors completing a fellowship, 29 (34%) completed a fellowship in pediatric otolaryngology. A logistic regression found that men have an increased odds of being on the Top Doctors list for more years than females (OR: 1.36, p < .001).

### Conclusion

The percentage of women named as Top Doctors was less than the proportion of women in otolaryngology. This may be attributed to gender differences we found in fellowship type and certification. Further research into the role of otolaryngology subspecialties in selection of Top Doctors is needed to better understand gender differences.

**Competing interests:** The authors have declared that no competing interests exist.

## Introduction

More women are becoming otolaryngologists. From 2005 to 2015, the percent of women otolaryngology residents increased from 25% to 36% [1]. In 2019, a cross-sectional analysis found that 24.5% of otolaryngology faculty were women [2]. However, despite the increase in the number of women otolaryngologists, gender imbalances within otolaryngology still exist. According to the Association of American Medical Colleges (AAMC), women represented 18.3% of active otolaryngologists in 2019 [3].

Previous studies have detailed the stark differences in career awards between men and women physicians. Atkinson *et al.* found that of 1,222 awards presented by 20 surgical societies, only 420 (25.6%) were awarded to women [4,5]. Additionally, a study of leadership positions of all United States otolaryngology societies found that only 23% of 160 studied leaders were women [6].

The Castle Connolly Top Doctor list is a directory of physicians from a variety of medical specialties. In order to become a Castle Connolly Top Doctor, one must be nominated by a fellow physician and selected by a physician-led research team [7]. Castle Connolly advertises its list as a tool for patients to find "trusted, quality care" and one study corroborated its usefulness in finding quality care in certain specialties [8]. However, it is not known if there are gender discrepancies in Top Doctor awardees. This study seeks to analyze the profiles of otolaryngologists selected by Castle Connolly's Top Doctor list in order to determine how these profiles compare to those of all otolaryngologists.

## Materials and methods

Top Doctor lists published in Castle Connolly-affiliated magazines were analyzed for Otolaryngology, Otolaryngology/Facial Plastic Surgery, and Pediatric Otolaryngology physicians [9]. Only the 2021 Top Doctor lists were analyzed. Magazines were excluded if they contained duplicate physicians that were already listed in another magazine or if they did not have an updated 2021 edition. All 39 magazines partnered with Castle Connolly were searched for a 2021 Top Doctor list that included Otolaryngology, Otolaryngology/Facial Plastic Surgery, and/or Pediatric Otolaryngology physicians. Twenty-seven of the 39 magazines were included in our study. This study was developed based on the methods of previous work that evaluated Castle Connolly's Top Doctors in urology lists [10].

Information on each physician was analyzed based on profiles on the Castle Connolly website. The physician's gender, degree (DO or MD), specialty, years in practice when given a Top Doctor award based on residency/fellowship graduation year, and the location where the physician practices based on the United States census region were recorded. If the physician held a faculty position (professor, associate professor, assistant professor, clinical professor, or clinical assistant professor), this was also recorded. Each physician profile on the Castle Connolly website contains milestones for how many years the physician has been recognized as a Top Doctor and we recorded this as 0–5 years, 5 years, 10 years, 15 years, or 20 years. We also collected data on any certifications that a physician had completed.

This study (STUDY00005617) was considered to be exempt by the University at Buffalo Institutional Review Board (IRB). This study is non-human research and did not require the use of reporting guidelines. Frequencies and comparisons were calculated when appropriate with SPSS 27 (Version 27.0.0.0, Armonk, NY: IBM Corp 2020). Kruskal Wallis, Fisher's Exact, or Chi-Square tests were used when appropriate. P-value < .05 was considered statistically significant.

**Table 1. Demographics of Castle Connolly 2021 Top Doctors in Otolaryngology, Otolaryngology/Facial Plastic Surgery, and pediatric otolaryngology.**

| Demographics | Gender | | Total | p-value |
|---|---|---|---|---|
| | Men N (%) | Women N (%) | | |
| Region | | | | < .88* |
| Northeast | 424 (85) | 74 (15) | 498 | |
| South | 132 (84) | 26 (17) | 158 | |
| Midwest | 139 (83) | 29 (17) | 168 | |
| West | 46 (85) | 8 (15) | 54 | |
| Specialty | | | | < .001* |
| Otolaryngology | 577 (85) | 101 (15) | 678 | |
| Otolaryngology/Facial Plastic Surgery | 109 (91) | 10 (8) | 119 | |
| Pediatric Otolaryngology | 56 (68) | 137 (32) | 82 | |
| Education | | | | .36** |
| MD | 736 (85) | 135 (16) | 871 | |
| DO | 6 (75) | 2 (25) | 8 | |

*Chi-squared test

**Fisher's exact test

## Results

Our analysis found 879 Top Doctors. Of the 879, 742 (84%) were men and 137 (16%) were women. Demographics of the physicians included can be seen in Table 1. There were 509 physicians who had completed a fellowship. Of the physicians that completed a fellowship, 85 (9.7% of the entire group) were women and 424 (48.2%) were men (Table 2). The number and genders of Top Doctors who had completed each fellowship can be seen in Table 2. The fellowship type varied significantly between genders (p = .002). Of the women Top Doctors completing a fellowship, 29 (34%) had completed a fellowship in pediatric otolaryngology.

Women had a significantly more recent graduation date and practiced for fewer years before being designated a Top Doctor than men (Table 3). A logistic regression found that men have an increased odds of being on the Top Doctors list for more years than women (OR: 1.36, p < .001) (Table 4). There was a significant difference between the number of years the person was recognized as a Top Doctor and gender as 93% of physicians recognized for twenty years were men. Three hundred and ninety Top Doctors in our study held a faculty position of whom 337 (86%) were men. Linear regression found that the physicians with more years on the Top Doctor list were more likely to have faculty positions (p < .001); however, there was no significant difference in education, fellowship, or certification. Additionally, the linear regression showed that Top Doctors with more years since graduation were less likely to have completed a fellowship (p < .001), but more likely to have a faculty position (p < .001).

## Discussion

Our study found that 16% of 2021 Top Doctors in Otolaryngology, Otolaryngology/Facial Plastics, and Pediatric Otolaryngology were women. In 2019, women represented 34.7% of residents and 18.3% of active otolaryngologists [3]. Our study found there are fewer women being recognized as a Top Doctor by Castle Connolly compared to the number of women otolaryngologists nationally. Furthermore, the fellowship types of the Top Doctors varied significantly by gender, with more women listed for pediatric otolaryngology than other subspecialties. An eight year study where candidates were surveyed at the American Board of

**Table 2. The number of Top Doctors that complete otolaryngology fellowships and gender of Top Doctors completing each fellowship.**

| Fellowship | Men N (%) | Women N (%) | Total N | P-value |
|---|---|---|---|---|
| Completed any fellowship | 424 (83) | 85 (17) | 509 | .29* |
| Facial Plastic & Reconstructive Surgery | 111 (91) | 11 (9) | 122 | .002** |
| Pediatric Otolaryngology | 67 (70) | 29 (30) | 96 | |
| Head and Neck Oncology | 77 (88) | 11 (12) | 88 | |
| Neurotology | 35 (80) | 9 (20) | 44 | |
| Laryngology | 23 (79) | 6 (21) | 29 | |
| Rhinology/Rhinoplasty | 8 (80) | 2 (20) | 10 | |
| Skull Base Surgery | 4 (57) | 3 (43) | 7 | |
| Otology | 6 (86) | 1 (14) | 7 | |

*Chi-squared test

**Fisher's exact test

Otolaryngology—Head and Neck Surgery oral examination found that women were significantly more likely to choose pediatric otolaryngology compared to men, with 30.9% of women and 15.8% of men choosing pediatrics [11]. Thus, the Top Doctor's fellowship gender distribution seems to match the gender distribution of the subspecialty within otolaryngology.

Linear regression found that the physicians with more years on the Top Doctor list were more likely to have faculty positions. Previous research has shown that women are underrepresented in academic medicine leadership. Uppal *et al*. found that being a man was significantly associated with having a full professorship title (P < .0001) [12]. Men and women have significantly different h-indexes (P < .0001) across all residency leadership positions [12]. Only 30% of residency program directors are women, which further supports that women are increasingly under-represented in leadership roles in otolaryngology [12]. Sulibhavi *et al*. in 2020 found that out of 90 otolaryngology residency programs analyzed, only four had department chairs who were women [13]. Further, Epperson *et al*. found that of the otolaryngology residency or fellowship directors, 26.3% of directors who were women were full professors while 53.6% of men were full professors [14]. Thus, previous research also demonstrates a significant difference between the genders in the percentage who have senior academic positions.

Additionally, our study found women had a significantly more recent graduation date and practiced for fewer years when recognized as a Top Doctor, compared to men. Thus, women are younger in their career than men when being recognized as Top Doctor. The lack of women on the Top Doctors list and in senior academic leadership positions may also be explained by the concept of the "leaky pipeline". The leaky pipeline theory describes the idea that at every step requiring decision making in an individual's career, more women are lost compared to men. For example, when choosing a medical specialty, women may be discouraged from applying to surgical fields due to a lack of role models or concern over being

**Table 3. Years in practice of top based on residency graduate year and years from graduation to Top Doctor designation, years shown as median [95% CI].**

| | Men | Women | P-value* |
|---|---|---|---|
| Residency graduation year | 1997 [1996–1998] | 2002 [1999–2004] | < .001 |
| Years from graduation to Top Doctor designation | 24 [23–25] | 19 [18–22] | < .001 |

*Kruskal Wallis test

**Table 4. Comparison of years physicians were recognized as a Top Doctor and physician certifications or faculty positions, stratified by gender.**

|  | Men (N%) | Women (N%) | p-value* |
|---|---|---|---|
| Years on Top Doctor List |  |  | < .001 |
| 0–5 years | 244 (78) | 69 (22) |  |
| 5 years | 197 (86) | 33 (14) |  |
| 10 years | 78 (85) | 14 (15) |  |
| 15 years | 94 (89) | 12 (11) |  |
| 20 years | 128 (93) | 9 (7) |  |
| Certification | 173 (91) | 18 (9) | .008 |
| Faculty position | 337 (86) | 53 (14) | .135 |

*Chi Squared test

accepted in a "male dominated" field [15,16]. Therefore, even though there are now more women than men matriculating into medical school, there are disproportionally fewer women in otolaryngology and even fewer holding academic senior positions. A study by Salem *et al.* found that women may leave academic medicine due to lack of mentoring, poor accommodations for parenting responsibilities, and/or perceived gender bias [17]. The lack of women in senior positions may also be due to women being promoted less often than men. Nocco and Larson found that women are promoted more slowly compared to men in academic medicine, even after controlling for hours worked, publications, tenure, and/or teaching awards [18].

Our study found that men have an increased odds of being on the Top Doctors list for more years than women. Previous research has found that women were less likely to receive other awards in the field of otolaryngology as well. Zambare *et al.* found that women received 28% of awards given by otolaryngology societies, women more often receiving a research or humanitarian award versus an achievement award [4]. In addition, Dossa *et al.* found that medical providers who were men were more likely to refer to surgeons who were men over women surgeons, even after adjustment for all other variables [19]. Therefore, since Castle Connolly uses peer nomination to choose Top Doctors, there may be an inherent bias towards men. This study and prior research shows women are recognized less often for their achievements, however, research has shown women surgeons have better outcomes. Wallis *et al.* found patients operated on by women surgeons were significantly less likely to die within 30 days when compared to patients operated on by men surgeons [20]. Additionally, Satkunasivam *et al.* found that increasing age was also associated with decreased post-operative mortality [21]. Therefore, the lack of women surgeons, especially older women surgeons, on the Top Doctors list does not appear to align with previous findings concerning postoperative outcomes.

Although we found an association between being a man and being on the Top Doctor list, prior research in orthopedic surgery has shown a significant increase in the percentage of women making the Top Doctors list from 2000 to 2020 [22]. Future research evaluating the distribution of women on Top Doctors lists over the span of multiple years may be of use.

## Limitations

Our study is limited to one year's listing of Top Doctors. The addition of more years of listings has the potential to change the gender proportions on the Top Doctor lists. Lastly, our research is limited because information was obtained from websites that may have outdated or

incorrect information about the physician. Multiple researchers collected data as well which could pose challenges to reporting data consistently.

The use of the Castle Connolly Top Doctor lists should not be assumed to reflect physician quality. To be nominated as a Top Doctor requires peer nomination and review of credentials by the Castle Connolly physician-led research team [7]. However, there is no previous research showing that the physicians listed on Top Doctor lists have better clinical outcomes compared to those not listed. The Top Doctors website does indicate that is does not allow physicians to pay to be featured on their website [23]. However, articles have stated that physicians can pay for an enhanced profile or buy a Top Doctors plaque [23]. This again leads us to question the motives behind the selection of Top Doctors. Furthermore, Top Doctors are featured in numerous magazines with which Castle Connolly has partnerships. John Connolly, founder of Castle Connolly, has stated that the Top Doctors issue is typically their number one seller both for advertising and on the newsstand [23]. This raises the question of a profit motive, both for Castle Connelly and for named physicians and for their nominators, who may be connected to them financially. However, the lack of women physicians recognized as Top Doctors is consistent with the gender distribution of all otolaryngology physicians and award distribution among men and women otolaryngologists should be further studied.

## Conclusion

Men have a higher odds of being on the Top Doctors list, and being on it for more years, than do women. The share of women awarded Top Doctor status was less than the proportion of women in otolaryngology nationally. Women on the Top Doctor list had a significantly more recent graduation date and practiced for fewer years compared to men. Top Doctor lists are created by profit-making ventures and how this contributes to selection is unknown. Future research should look into why there is a lack of senior women being recognized for achievements when women surgeons are having statistically less mortality post-operatively.

## Author Contributions

**Conceptualization:** Lauren A. DiNardo, Alyssa D. Reese, Michele M. Carr.

**Data curation:** Maya Raghavan, Meagan Sullivan.

**Formal analysis:** Alyssa D. Reese, Maya Raghavan, Michele M. Carr.

**Investigation:** Lauren A. DiNardo, Alyssa D. Reese.

**Methodology:** Lauren A. DiNardo, Alyssa D. Reese, Michele M. Carr.

**Resources:** Michele M. Carr.

**Supervision:** Lauren A. DiNardo, Michele M. Carr.

**Validation:** Michele M. Carr.

**Writing – original draft:** Lauren A. DiNardo, Alyssa D. Reese, Maya Raghavan, Meagan Sullivan, Michele M. Carr.

**Writing – review & editing:** Lauren A. DiNardo, Alyssa D. Reese, Maya Raghavan, Meagan Sullivan, Michele M. Carr.

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
