## [Decision Letter · Decision Letter 0]

4 Mar 2024

Gender Distribution of Top Doctors in Otolaryngology-Head and Neck Surgery

PONE-D-24-04095

Dear Dr. DiNardo,

We’re pleased to inform you that your manuscript has been judged scientifically suitable for publication and will be formally accepted for publication once it meets all outstanding technical requirements.

Kind regards,

Jeyasakthy Saniasiaya, MD, MMed ORLHNS, FEBORLHNS

Academic Editor

PLOS ONE

Additional Editor Comments (optional):

Reviewers' comments:

Reviewer's Responses to Questions

**Comments to the Author**

1. Is the manuscript technically sound, and do the data support the conclusions?

Reviewer #1: Yes

Reviewer #2: Yes

2. Has the statistical analysis been performed appropriately and rigorously? 

Reviewer #1: Yes

Reviewer #2: Yes

3. Have the authors made all data underlying the findings in their manuscript fully available?

Reviewer #1: Yes

Reviewer #2: Yes

4. Is the manuscript presented in an intelligible fashion and written in standard English?

Reviewer #1: Yes

Reviewer #2: Yes

5. Review Comments to the Author

Reviewer #1: Well- written and timely paper. The data us well presented and the review process and statistical analysis are solid. This is interesting data for the readership. This information has pertinence as it is timely and represents a significant gap that is occurring in all fields of medicine.

Reviewer #2: Thank you for your interesting article reviewing the gender distribution on the Top Doctor list in OHNS. This paper adds to the current literature demonstrating gender inequity in the field of otolaryngology (eg. award recipient, leadership position, faculty position, senior authorship, journal editorial boards).

The research question and methodology are sound and statistical analyses were appropriate.

This study collected and analyzed data from the 2021 Castle Connolly affiliated magazine Top Doctor list. It examined proportion of female vs male otolaryngology MDs and different variables associated with the profiles. The study found that the 2021 Top Doctor list still lags behind the number of female otolaryngologists in the US. This paper also highlighted the fact that Top Doctors uses peer nomination to choose their list, which may be an inherent bias which is important for readers/patients to be aware of.

I agree that one of the limitation of this study is the analysis of only 1 year of the Top Doctor lists but it would be prudent to expand this evluation over multiple years to examine the trend in OHNS. I hope to see future research add to this important body of work.

6. PLOS authors have the option to publish the peer review history of their article (what does this mean?). If published, this will include your full peer review and any attached files.

Reviewer #1: No

Reviewer #2: No

---

## [Editor Report · Acceptance letter]

5 Apr 2024

PONE-D-24-04095 

PLOS ONE

Dear Dr. DiNardo, 

I'm pleased to inform you that your manuscript has been deemed suitable for publication in PLOS ONE. Congratulations! Your manuscript is now being handed over to our production team.

Kind regards, 

on behalf of

Dr. Jeyasakthy Saniasiaya 

Academic Editor

PLOS ONE